# Sensing Features of the Fano Resonance in an MIM Waveguide Coupled with an Elliptical Ring Resonant Cavity

**Hao Su** [1,2], **Shubin Yan** [1,2,3,*], **Xiaoyu Yang** [1,2], **Jing Guo** [2], **Jinxi Wang** [1,2] and **Ertian Hua** [2]

1    School of Instrument and Electronics, North University of China, Taiyuan 030051, China;
     suh@zjweu.edu.cn (H.S.); yangxy@zjweu.edu.cn (X.Y.); wangjx@zjweu.edu.cn (J.W.)
2    School of Electrical Engineering, Zhejiang University of Water Resources and Electric Power,
     Key Laboratory for Technology in Rural Water Management of Zhejiang Province;
     Hangzhou 310018, China; guojing@zjweu.edu.cn (J.G.); het@zjut.edu.cn (E.H.)
3    School of Electrical and Control Engineering, North University of China, Taiyuan 030051, China
*    Correspondence: yanshb@zjweu.edu.cn; Tel.: +86-186-3611-2255



**Featured Application: The values of the refractive index have a linear relationship with the concentration of hemoglobin at all wavelengths. By means of measuring the refractive index of the blood group, researchers can calculate the concentration of hemoglobin (g/L) within the erythrocytes, which is a vital parameter to detect the blood health condition. The Fano resonance relies on its advantage that it is sensitive to the changes of refractive indices, making it become a promising platform for designing highly integrated medical optical sensors to detect concentration of hemoglobin and monitor body health.**

**Abstract:** In this article, a novel refractive index sensor composed of a metal–insulator–metal (MIM) waveguide with two rectangular stubs coupled with an elliptical ring resonator is proposed, the geometric parameters of which are controlled at a few hundreds of nanometer size. The transmission feature of the structure was studied by the finite element method based on electronic design automation (EDA) software COMSOL Multiphysics 5.4 (Stockholm, Sweden). The rectangular stub resonator can be thought of as a Fabry–Perot (FP) cavity, which can facilitate the Fano resonance. The simulation results reveal that the structure has a symmetric Lorentzian resonance, as well as an ultrasharp and asymmetrical Fano resonance. By adjusting the geometrical parameters, the sensitivity and figure of merit (FOM) of the structure can be optimized flexibly. After adjustments and optimization, the maximum sensitivity can reach up to 1550 nm/RIU (nanometer/Refractive Index Unit) and its FOM is 43.05. This structure presented in this article also has a promising application in highly integrated medical optical sensors to detect the concentration of hemoglobin and monitor body health.

**Keywords:** surface plasmon polaritons; refractive index sensor; sensitivity; Fano resonance; metal–insulator–metal waveguide

## 1. Introduction

As a result of the advantage of overcoming the limit of light diffraction, surface plasmon polaritons (SPPs) have become a promising choice to implement high-density integrated optical circuits, making more and more researchers lay the great emphasis on metal–insulator–metal (MIM) structure waveguides based on SPPs [1–3]. SPP is a phenomenon in which incident photons are coupled with electrons that are free in the surface of the conductor [4,5]. In recent years, researchers have proposed

a large number of devices related to MIM waveguides to realize a variety of instruments that have different functions, such as filters [6–11], splitters [12,13], sensors [14–21], and they greatly promote the development of novel instruments.

Compared with Lorentzian resonance, Fano resonance has been applied widely in sensing instruments because of its ultrasharp and asymmetrical transmission spectrum, which is sensitive to the changes of the structure's parameters and the surrounding environment [22–26]. There are two vital parameters that evaluate the sensing performance of the system: the sensitivity S and the figure of merit (FOM). Domestic and foreign scholars have recently proposed and studied a variety of sensing structures that make full use of the sensitivity of the Fano resonance to realize different applications. Rukhsar Zafar et al. [27] designed an MIM waveguide coupled with a pair of elliptical ring resonators; the sensitivity can reach up to 1100 nm/RIU and its FOM is 224. A compact refractive index nanosensor composed of an MIM waveguide coupled with a circular split-ring resonant cavity was proposed by Yang et al. [28], whose sensitivity is up to 1500 nm/RIU and the FOM can reach 65.2. Chen [29] proposed a structure composed of an MIM waveguide and rectangular side-coupled cavity, whose sensitivity is 1280 nm/RIU. Chen also found that by means of increasing the number of rectangular resonance cavities, researchers can get more Fano resonance peaks. The FOM of the M-type resonant cavity structure designed by Qiao [30] et al. is up to $1.56 \times 10^5$, but its sensitivity is only 780 nm/RIU.

In this article, a structure composed of an MIM, two symmetric rectangular stubs and the elliptical ring resonator, is investigated. The rectangular stub resonator can be thought as a Fabry–Perot (FP) cavity, which can facilitate the Fano resonance. The sensitivity and FOM are studied by means of trying to alter different geometrical parameters of the system. According to numerous simulations, the proposed system can serve as a highly efficient refractive index sensor with a sensitivity of 1550 nm/RIU and FOM of 43.05. Zhernovaya et al. [31] proposed that the measured values of the refractive index have a linear relationship with the concentration of hemoglobin at all wavelengths, and showed a linear relationship between them. Herein, the Fano resonance relies on its advantage that it is sensitive to the changes of refractive indices, making it become a promising platform for designing highly integrated medical optical sensors to detect the concentration of hemoglobin and monitor body health.

However, due to the restriction of the etching precision [32,33], it is difficult to control the coupling distance $d$ and fabricate the MIM waveguide-coupled resonator system. As a result, this article mainly provides a theoretical analysis for future experimental studies.

## 2. Structure Design and Theoretical Analysis

Figure 1 shows the two-dimensional schematic of the structure proposed in this paper, in which the white and green are separate dielectric air and metal Ag layers. To be specific, the structure is composed of an MIM waveguide with two rectangular stubs and an elliptical ring resonator, with the following defined parameters: $R_1$ and $R_2$ are the length of the long axis and the length of the short axis of the outer elliptical resonator, respectively; $r_1$ and $r_2$ are the length of the long axis and the length of the short axis of the inner elliptical resonator taken separately; $d$ is the coupling distance between the elliptical ring resonator and MIM waveguide; $h$ represents the height of two rectangular stubs; $l$ is the distance between two symmetric rectangular stubs; and $w$ is the width of the MIM waveguide. Compared with the circular ring resonator ($R_1 = R_2$), the elliptical one ($R_1 \neq R_2$) can obtain greater optical performance. The relative dielectric constant of Ag can be expressed by the model of Drude (1) [34]:

$$\varepsilon_d(\omega) = \epsilon_\infty - \frac{\omega_p^2}{\omega(\omega + i\omega\gamma)} \tag{1}$$

where $\omega$ is the angular frequency of the incident wave which is in a vacuum, the infinite dielectric constant $\epsilon_\infty = 3.7$, the plasma oscillation angular frequency $\omega_p = 9.1eV(1.38 \times 10^{16} \text{ rad/sec})$, and the electron collision frequency $\gamma = 0.018eV(2.73 \times 10^{13} \text{ rad/sec})$.

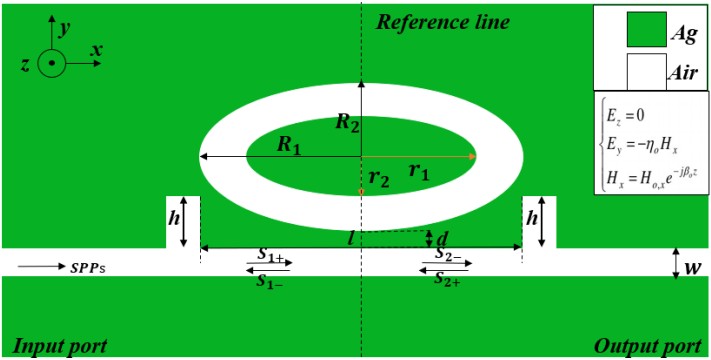

**Figure 1.** The 2D schematic of the structure composed of a metal–insulator–metal (MIM) with two symmetric rectangular stubs coupled with the elliptical ring resonant cavity. The expression of the incident electric and magnetic field is shown in the upper right corner of the picture.

The size of the simulation domain is $1200 \times 900$ nm. The width of the MIM waveguide is set at $w = 50$ nm and at this time the MIM waveguide only exists the fundamental transverse magnetic (TM$_0$). When the transverse magnetic mode is TM$_0$, it helps SPP waves to propagate [35]. Perfectly matched layers (PMLs) are set at the top and bottom boundaries of the structure. For directly understanding the transmission features of the structure, we make full use of the EDA software COMSOL Multiphysics 5.4 [36], and with the help of its finite element method (FEM), we can observe the transmission feature of the structure. The mesh is ultrafine in order to ensure the accuracy of calculations. The light imports from the input port $P_1$, then propagates in the MIM waveguide, and eventually exports at the output port $P_2$. The transmittance is defined as $T = (S_{21})^2$, where $S_{21}$ is the transmission coefficient from input port $P_1$ to output port $P_2$. The interval step of incident wavelength is set to 5 nm and the pattern of the incident light is TM.

## 3. Simulations and Results

As shown in Figure 2, the red curve indicates the transmission spectrum of the structure without two rectangular stubs, the blue one describes the transmission spectrum of the structure without the elliptical ring resonator, the green one shows the transmission spectrum of the structure with only one rectangular stub, while the black one shows the whole structure's transmission spectrum. The parameters of the structure are $R_1 = 300$ nm, $R_2 = 140$ nm, $r_1 = 150$ nm, $r_2 = 70$ nm, $d = 10$ nm, $l = 480$ nm, $h = 60$ nm.. When the wavelengths are 1105 and 1545.5 nm, the transmission spectrum of the black curve decreases sharply, and two narrow resonance lines are formed. Among them, the left dip is a symmetric Lorentzian resonance, while the right one is a typically asymmetric Fano resonance. In general, when the continuous broadband state and the discrete narrowband state interact with each other, a Fano resonance will be generated [37,38]. When there are only two rectangular stubs and no elliptical ring resonator in the structure, a broad continuous spectrum with relatively high transmittance is formed. The transmission spectrum of the single elliptical ring resonator has shapes of the Lorentzian resonance and Fano resonance, which are considered as discrete narrowband states. When both two rectangular stubs and the elliptical ring resonator exist in the meantime, the coupling of two discrete narrowband states and a broad continuous state will generate a symmetric Lorentzian resonance, as well as an ultrasharp and asymmetrical Fano resonance in the system, as is shown in Figure 2. In this article, we mainly study the characteristics of the Fano resonance.

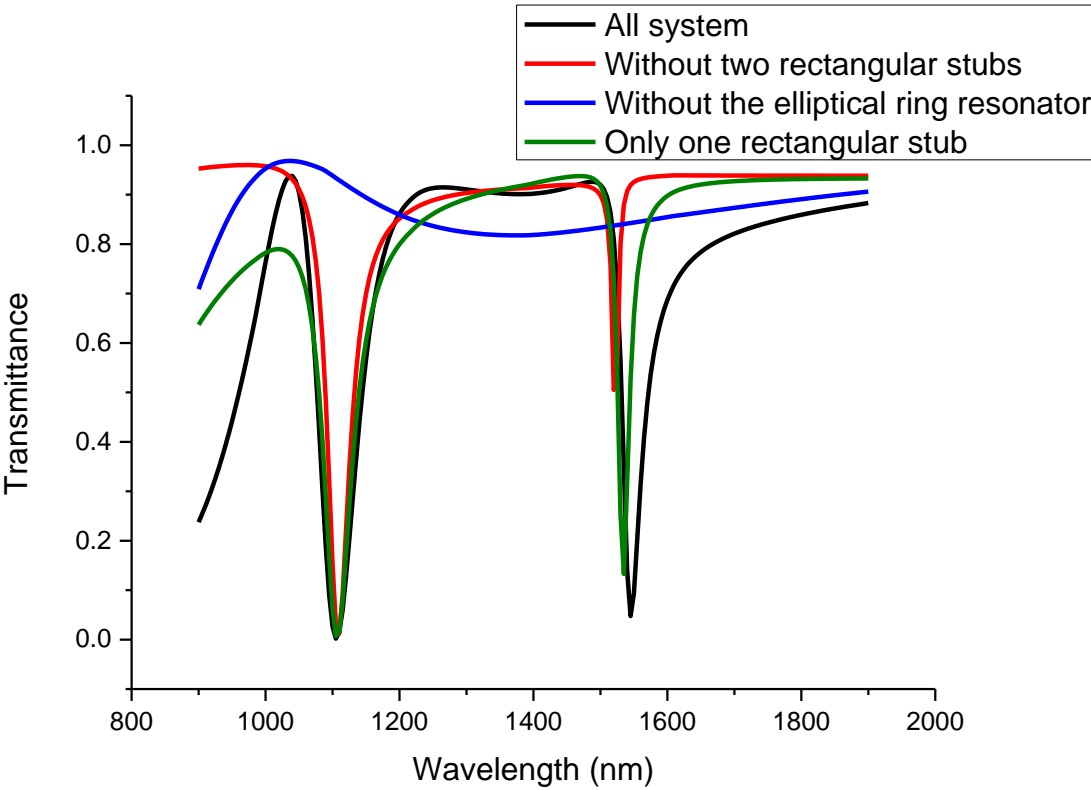

**Figure 2.** Transmission spectrum of the whole system (the black line), without two rectangular stubs (the red line), without the elliptical ring resonator (the blue line), with only one rectangular stub (the green line) when $R_1 = 300\,\text{nm}$, $R_2 = 140\,\text{nm}$, $r_1 = 150\,\text{nm}$, $r_2 = 70\,\text{nm}$, $d = 10\,\text{nm}$, $l = 480\,\text{nm}$, $h = 60\,\text{nm}$.

To deeply understand the principle of the Fano resonance, we study normalized Hz field distributions of the MIM waveguide coupled with the single elliptical ring resonator, the MIM waveguide with single two rectangular stubs, as well as the entire structure at the resonance dip. When $\lambda = 1105\,\text{nm}$, a strong Hz field appeared in the left rectangular stub and elliptical ring resonator (as shown in Figure 3a). The field distribution of the elliptical ring resonator is symmetric on the reference line. SPPs are confined in the left stubs and elliptical ring resonator because of the destructive interference of the two excitation pathways. The two pathways include a broad resonance spectrum from the rectangular stub resonator and a narrow one from the elliptical ring resonator. As a result, the SPPs do not transmit to the other side of the MIM waveguide. When $\lambda = 1545.5\,\text{nm}$, the SPPs scarcely resonate in the rectangular stub, which shows that the incident SPPs and the SPPs which escape from the stub resonator into the MIM waveguide have an interaction. That is to say, at this time a coherence enhancement occurs in the MIM waveguide. Herein, the rectangular stub resonator can be thought as an FP cavity. The FP cavity is composed of an air dielectric layer sandwiched between two silver dielectric layers. The FP cavity forms a grating structure. The structure makes sure that the incident light diffracts in order to excite the propagation mode in the waveguide. As a result, the interference between the diffracted light and the directly transmitted light is more likely to happen. It accounts for the phenomenon that the SPPs scarcely resonate in the rectangular stub. Compared with Figure 3d, it is obvious that when two rectangular stubs are added into the structure, a stronger resonance generates in the elliptical ring resonator, which can facilitate the Fano resonance. Meanwhile, when a strong resonance is generated, the SPPs are nearly limited in the elliptical ring resonator. As a consequence, the structure has a relatively low transmittance at the dip.

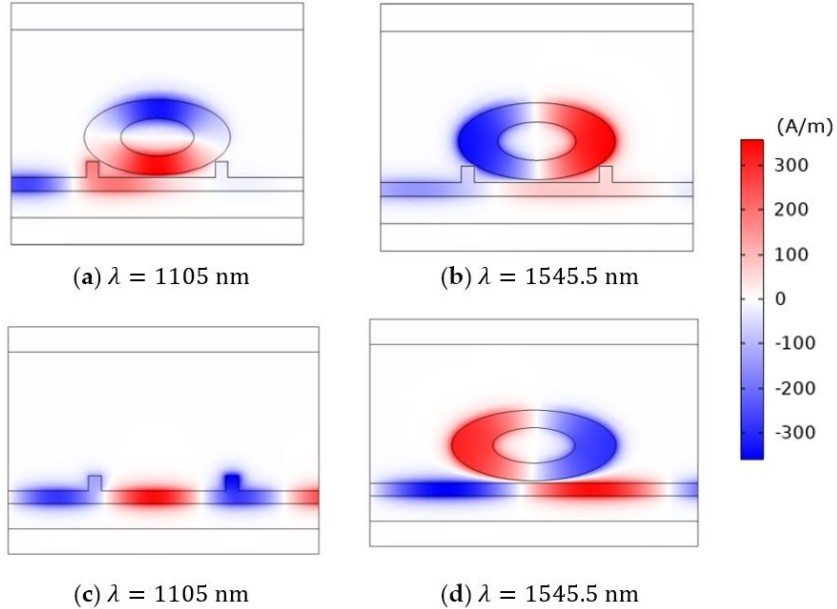

**Figure 3.** The Hz field distribution of the resonant dip: (**a**) the whole system at $\lambda = 1105$ nm; (**b**) the whole system at $\lambda = 1545.5$ nm; (**c**) without the elliptical ring resonator at $\lambda = 1105$ nm; (**d**) without two rectangular stubs at $\lambda = 1105$ nm.

The Fano resonance is easily influenced by changing refractive indices of the dielectric due to its asymmetrical and sharp line shape. Therefore, we can make use of the sensitivity of the Fano resonance to monitor the changes of vital parameters. There are two vital parameters for evaluating the sensing performance of the Fano system: the sensitivity S and the FOM, which are expressed as (2) and (3):

$$S = \frac{\Delta\lambda}{\Delta n} \tag{2}$$

$$FOM = \frac{S}{FWHM} \tag{3}$$

Herein, *FWHM* means the full width at the half maximum. Figure 4a describes the transmission spectrum of the system when the refractive index changes from 1 to 1.05 at an interval of 0.01. When the refractive index changes, other parameters of the structure remain the same, as shown in Figure 3a. The fitting line shows that the sensitivity is 1550 nm/RIU with $R_1 = 300$ nm, $R_2 = 140$ nm, $r_1 = 150$ nm, $r_2 = 70$ nm, $d = 10$ nm, $l = 480$ nm, $h = 60$ nm., and its FOM is 43.05.

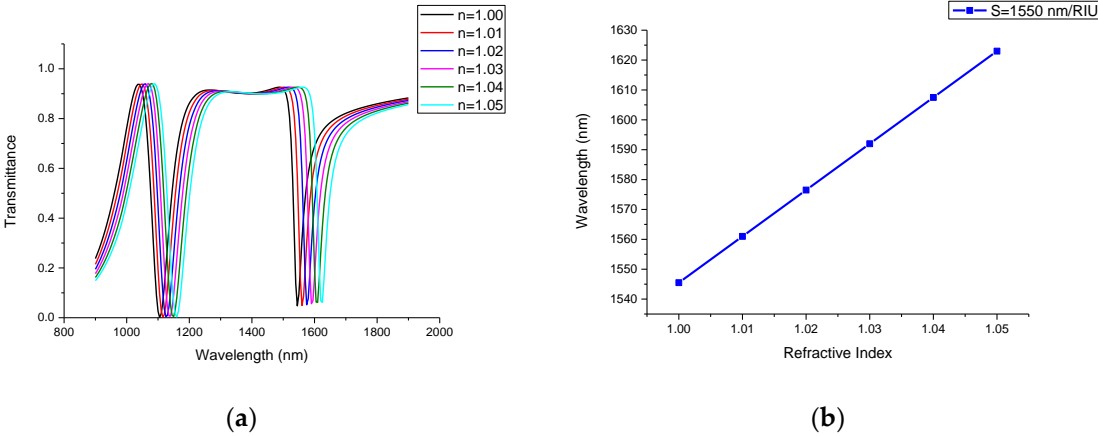

**Figure 4.** (**a**) Transmission spectrum of different refractive indices; (**b**) the fitting line of the wavelength of the Fano resonance dip with the changes of refractive index.

In this part, we mainly study the influence on the transmission spectrum when we change several main parameters of the structure. First of all, when the influence of elliptical long axis $R_1$ is studied, $R_1$ increases from 260 to 300 nm, and other parameters, such as $R_2 = 140$ nm, $r_1 = 150$ nm, $r_2 = 70$ nm, $d = 10$ nm, $l = 480$ nm, $h = 60$ nm are kept constant. In addition, the proportional relationship between $R_1$ and $r_1$ is two at this time. For an MIM waveguide coupling resonator, the resonance wavelength can be calculated by standing wave theory:

$$\lambda_m = \frac{2Re(n_{eff})L}{m - \psi_r/\pi} \tag{4}$$

where $L$ is the length of the cavity, positive integer $m$ is the number of antinodes of the standing SPP wave, $\psi_r$ is the phase shift of the beam reflected at one end of the cavity, and $Re(n_{eff})$ is the real part of the effective index of an MIM waveguide. From the results of simulation shown in Figure 5a, we can find that as $R_1$ increases, the Fano resonance is red-shifted, which is consistent with Equation (4). That is, $\lambda_m$ increases when $L$ increases. Here, the increase of $L$ is achieved by increasing $R_1$. Meanwhile, when $R_1$ increases, the transmittance of resonance dip on the left dip increases slightly, while the right one shows an obvious decrease that is desired. From the fitting lines of various $R_1$ in Figure 5b, we can find that with the increase of $R_1$, sensitivities of the structure become better. Theoretically, we can get the greater sensitivity by increasing $R_1$ continuously. However, the increase of $R_1$ is not conducive to the miniaturization of the cavity area. At the same time, in order to achieve a low transmittance, $R_1$ is not less than 290 nm.

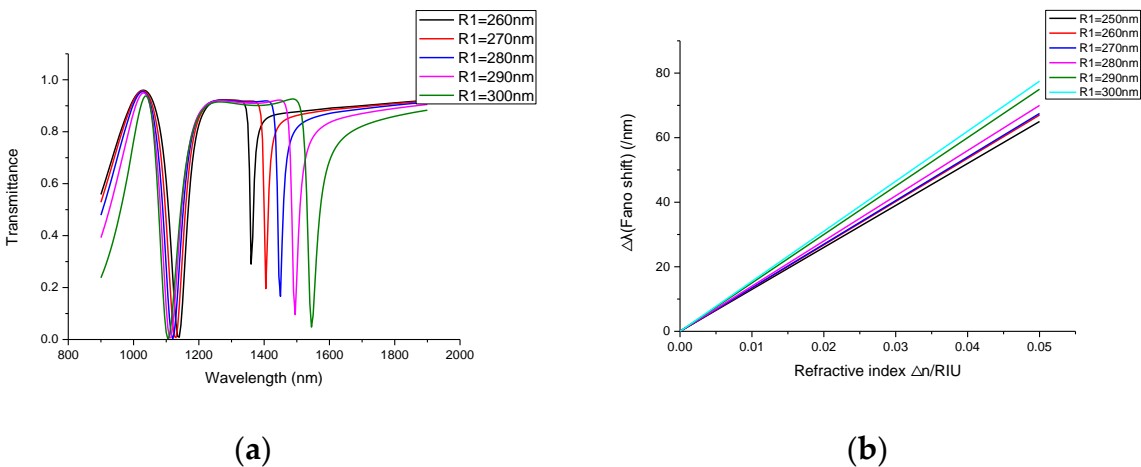

**(a)**                                                      **(b)**

**Figure 5.** (**a**) Transmission spectrum of different elliptical long axis $R_1$; (**b**) fitting line of the wavelength of the Fano resonance dip with the changes of elliptical long axis $R_1$.

In addition, we also study the influence of the coupling distance $d$ on the dip of Fano resonance. In the series of this group's simulation results, as is shown in Figure 6a, when $d$ increases from 10 to 30 nm at the interval of 5 nm, the Fano resonance is blue-shifted and the transmittance of the resonance dip increases. When $d = 5$ nm, due to rectangular stubs being tangent to the elliptical ring resonator, the Fano resonance cannot be generated and we set the minimum value of $d$ to be 10. According to this finding, we find that the coupling distance $d$ between the elliptical ring resonator and the MIM waveguide has a significant influence on the transmittance of the Fano resonance dip. In the following parts, with the aim of getting a low transmittance, the coupling distance $d$ should be small, so we keep $d = 10$ nm constant. In general, the lower transmittance can result in a larger extinction ratio and a smaller FWHM, which is useful for obtaining a high FOM and good sensing resolution.

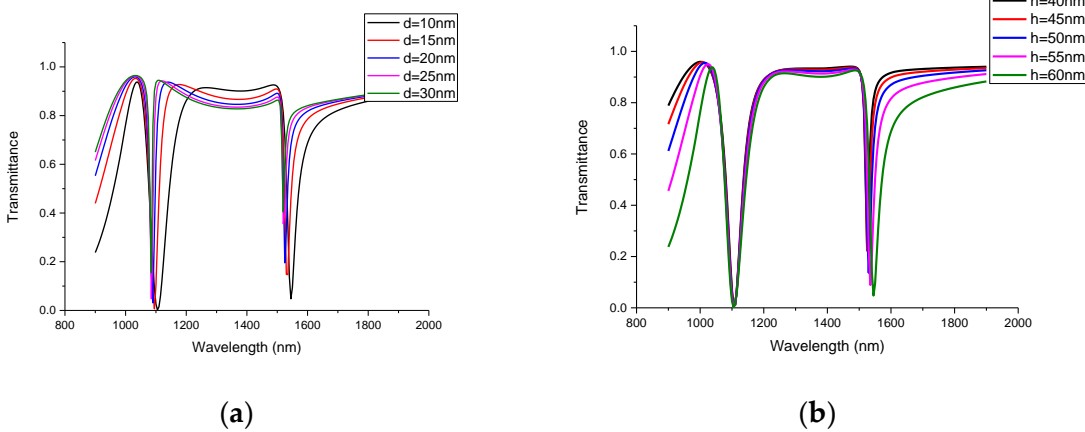

(**a**)                                   (**b**)

**Figure 6.** (**a**) Transmission spectrum of different coupling distances *d*; (**b**) transmission spectrum of different heights of rectangular stubs *h*.

We also try to change the parameter *h*, which represents heights of two rectangular stubs. As shown in Figure 6b, when *h* changes from 40 to 60 nm, the left dip nearly stays the same, while the right one decreases slightly and the FWHM of the transmission spectrum becomes broader. According to the previous analysis, a smaller FWHM means a higher FOM. We make a line chart to record the FOM of different heights of rectangular stubs depending on the results of the simulation. As vividly shown in Figure 7, when *h* increases from 40 to 60 nm at the interval of 5 nm, FOM of the structure decreases from 151.96 to 43.05. However, compared with *h* = 60 nm, the sensitivity of *h* = 55 nm decreases to 1500 nm/RIU and the transmittance becomes higher at the same time. Therefore, it is necessary to compromise between the sensitivity and FWHM.

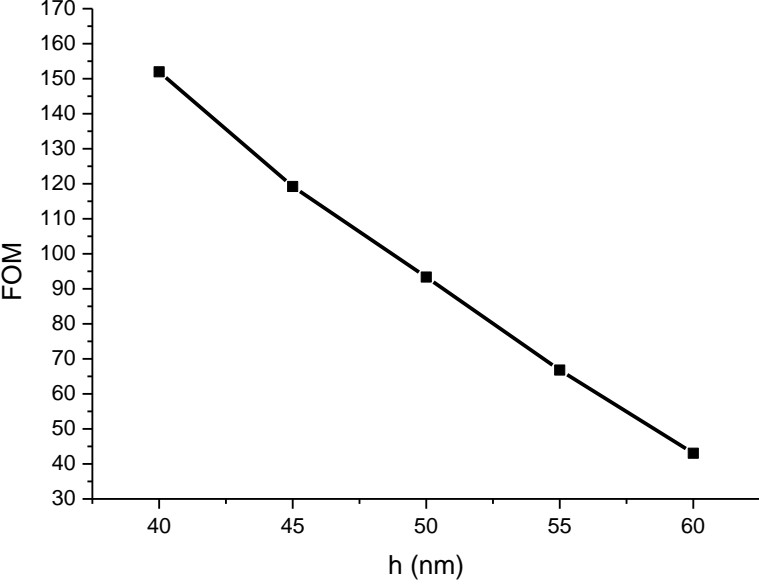

**Figure 7.** The values of FOM when heights of the two rectangular stubs increase from 40 to 60 nm.

Finally, we also study the influence of the other parameters $R_2, r_1, r_2$ and *w* on the properties of the structure. As shown in Figure 8a, when changing the length of the long axis of the inner elliptical resonator $r_1$ from 130 to 170 nm at the interval of 10 nm, we can find that the Lorentzian resonance and Fano resonance are red-shifted. However, the impact of $r_1$ on the Fano resonance is small compared with the Lorentzian resonance. Then, we change the length of the short axis of the inner elliptical resonator $r_2$ from 50 to 90 nm at the interval of 10 nm. The simulation result of different lengths of the

short axis of the inner elliptical resonator is shown in Figure 8b. It is found that the dip wavelength of the Lorentzian resonance is nearly unchanged, while the Fano resonance is red-shifted. At the same time, effects of $R_2$ and $w$ on the properties of the structure are also studied, as shown in Figure 8c,d. When the length of the short axis of the outer elliptical resonator becomes larger, the dip wavelength of Fano resonance is blue-shifted, the FWHM gets broader, and the transmittance of the Fano resonance dip becomes lower, which can be explained by the fact that when $R_2$ changes from 130 to 145 nm, the coupling distance between the elliptical ring resonator and MIM waveguide gets smaller and the coupling strength becomes stronger. In addition, when the width of the MIM waveguide gets larger, the dip of the Fano resonance is nearly unchanged. At the same time, in order to ensure that the MIM waveguide only exists the fundamental transverse magnetic ($TM_0$), the reasonable value of $w$ is around 50 nm.

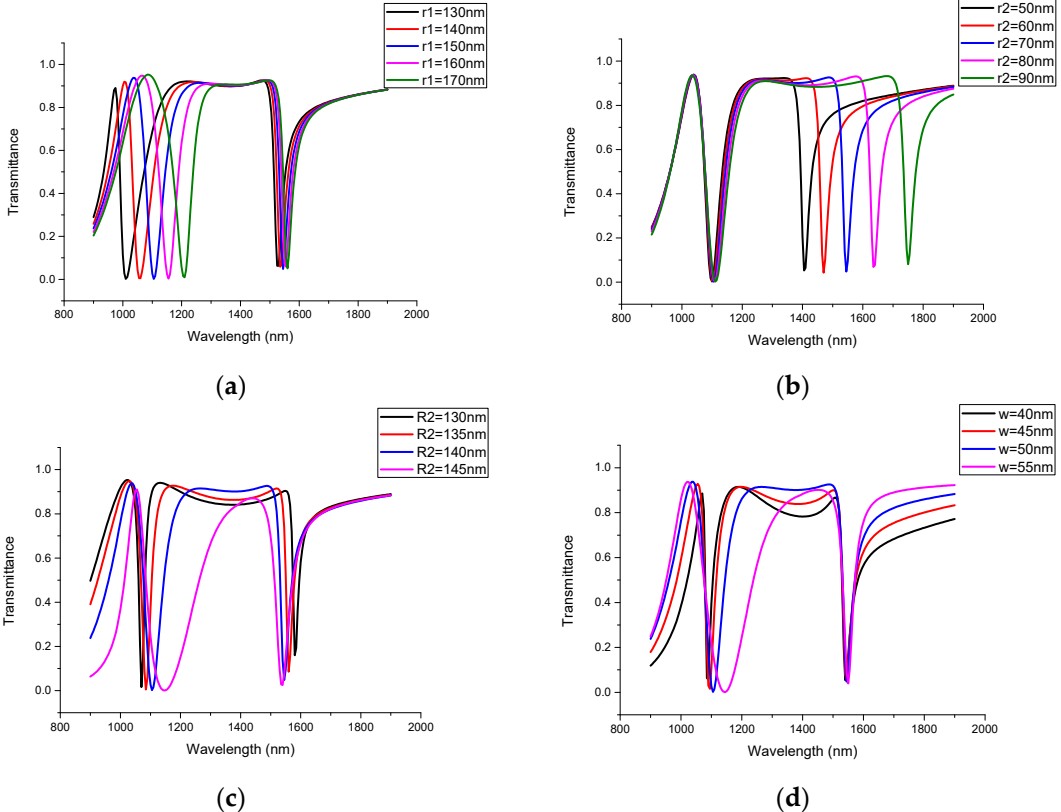

**Figure 8.** (**a**) Transmission spectrum of different lengths of the long axis of the inner elliptical resonator $r_1$; (**b**) Transmission spectrum of different lengths of the short axis of the inner elliptical resonator $r_2$; (**c**) Transmission spectrum of different lengths of the short axis of the outer elliptical resonator $R_2$; (**d**) Transmission spectrum of different widths of the MIM waveguide $w$.

Combined with the above analyses, we conclude that the wavelength of the Fano resonance mainly relies on the relevant parameters of the elliptical ring resonator, such as $R_1$ and $r_2$. Furthermore, it is found that when we change the parameters related to the MIM waveguide or the coupling distance, such as $d$, $h$, and $R_2$, the transmittance and FOM of the Fano resonance will be altered at the same time, which could be explained by the phenomenon that the SPPs are almost confined within the elliptical ring resonator and few SPPs are in the MIM waveguide.

## 4. Applications in Medical Detection

Currently, medical instruments that measure somatic function attract great interest of companies and individuals. The structure presented in this article is suitable for the detection of health indicators that are sensitive to refractive index. We can use the concentration of dry hemoglobin (g/L) at

temperature T (°C) to detect our health condition. According to the Barer's study, the refractive index has a linear relationship with C [39], which is depicted in Equation (5):

$$n = n_0 + \xi C \tag{5}$$

where $n_0$ is the refractive index when the concentration of hemoglobin (several nanometers) C is 0 and $\xi$ is the specific refraction increment. According to the study by Zhernovaya et al. [35], the measured values of the refractive index have a linear relationship with the concentration of dry hemoglobin at all wavelengths. In addition, the wavelength of the Fano resonance dip also has a linear relationship with the refractive index. Therefore, by means of measuring the wavelength of the Fano resonance of the blood group, researchers can calculate the refractive index and then observe C within the erythrocytes, which is a vital parameter to detect the blood health condition. In using the proposed structure as a health detection sensor, the blood sample is placed in the MIM waveguide. We can pass the blood sample through the filter to remove large cells before we place the blood in the waveguide. Then, researchers can calculate the concentration within the erythrocytes to detect the health condition of the patient. The Fano resonance relies on its advantage that it is sensitive to the changes of refractive indices, making it become a promising platform for designing highly integrated medical optical sensors to detect concentration of hemoglobin and monitor body health.

## 5. Conclusions

In this article, a refractive index sensor is proposed which is composed of an MIM waveguide, two rectangular stubs, and an elliptical ring resonator. The process of simulation was implemented by using the finite element method based on the EDA software COMSOL Multiphysics 5.4. The results of simulation reveal that the structure has a symmetric Lorentzian resonance, as well as an ultrasharp and asymmetrical Fano resonance. By adjusting the geometrical parameters, the sensitivity and FOM of the structure can be optimized flexibly. By comparing the results of simulation, we find that the wavelength of the Fano resonance mainly relies on the relevant parameter of the elliptical ring resonator, such as $R_1$ and $r_2$. Furthermore, it is found that when we change the parameter related to the MIM waveguide or the coupling distance, such as $d$, $h$, and $R_2$, the transmittance and FOM of the Fano resonance will be altered at the same time, which could be explained by the phenomenon that the SPPs are almost confined within the elliptical ring resonator and few SPPs are in the MIM waveguide. After adjustments and optimization, the maximum sensitivity can reach up to 1550 nm/RIU with a FOM of 43.05. The structure presented in this article has a promising application in highly integrated medical optical sensors to detect concentration of hemoglobin and monitor body health.

**Author Contributions:** Conceptualization, S.Y. and H.S.; software, H.S.; validation, S.Y. and X.Y.; formal analysis, J.G.; investigation, J.W.; data curation, E.H.; writing—original draft preparation, H.S.; writing—review and editing, S.Y., X.Y, and J.W.; supervision, E.H. All authors have read and agreed to the published version of the manuscript.

**Funding:** This work was supported by the National Natural Science Foundation of China (Grant No. 61675185,61875250,61975189) and sponsored by the Fund for Shanxi '1331 Project' Key Subject Construction.

**Conflicts of Interest:** The authors declare no conflict of interest.

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
