# Peer review of "Sensing Features of the Fano Resonance in an MIM Waveguide Coupled with an Elliptical Ring Resonant Cavity"

_applsci, doi:10.3390/app10155096_

Round 1

Reviewer 1 Report

This manuscript describes the theoretical structural optimization of MIM waveguide of two rectangular and an elliptical ring resonator for Fano resonance-based refractive index sensors and their diagnosis techniques. This manuscript orderly describes the calculation results and the mechanisms and thus, the reviewer thinks that the results described in this manuscript are reasonable. In addition, the authors demonstrated that these structures, which exhibit a high FoM, are applicable to the biosensing such as hemoglobin detection. Therefore, the reviewer thinks that this manuscript is suitable for the publication in applied sciences.

Author Response

Thank you for your affirmation of my research work. The revised manuscript has been uploaded. I sincerely hope to get your precious opinions again.

Reviewer 2 Report

In this manuscript, the authors numerically design and proposed a novel refractive index sensor comprised of a metal-insulator-metal (MIM) waveguide with two rectangular stubs coupled with an elliptical ring resonator using FEM. They claim that a symmetric Lorentzian resonance as well as an ultra-sharp and asymmetrical Fano resonance. Technically, the authors employ the numerical method (COMSOL Multiphysics) with apparent authority and the results seem valid and are interested to the readers. However, the necessary physical mechanism is absent in the manuscript to explain the results. In my opinion, a major revision is needed to accommodate the high quality requirements of the Journal.

Q1. The structural parameters, h, R1, R2, r1, r2, â„“ and d should be mentioned in the text and figure caption of figure 2.

Q2. if possible, please briefly elucidate in the text "What is the difference of the optical performance of elliptical ring resonator (R1≠R2) and the circular one (R1=R2)?

Q3. More information regarding Eq. (5) should be given. (e.g., ? should have a definition).

Q4. The color scale shown in Figure 3 (3x10^2 x10^2) is a wrong one. The unit of magnetic intensity (A/m) should be given in the color scale of figure 3.

Q5. Please briefly discuss the results if d=5 nm is included in figure 6.

Q6. The structural parameter h used in figure 3 is h=110 nm. However, the influence of transmittance specturm on h investigated in figure 6 is in the range of 40-60 nm. Why?

Q7. Line 157-158 in page 5, it is written that " ... we can find that as the increase of R1, the Fano resonance is red-shifted." Please describe the mechanism in more detail (or quote the relevant literature) why R1 have an impact influence on the transmittance spectrum.

Q8. Is there any limitation or threshold of the value of R1? Please give an available range of R1 (if possible).

Q9. For the simulation issue  

  • In order to be beneficial for the reader to follow this work, the version of COMSOL software used in this work should be given in the text and the website of software should be cited in the reference.
  • Is the simulation model a 2D or 3D model? I think it is a 2D one. Please describe in the text.

(3) The size of 2D (x- and y- axis) simulation models of the proposed structure, including the size of simulation domain and the location of perfectly matched layer (PML), mesh number, interval step of incident wavelength, and the pattern of the incident light, should be mentioned in the text.

(4) The notation of the incident electric and magnetic field directions and wave vector direction should be indicated in the inset of Fig. 1.

(5) What is the aberration of "EDA"? Please give a definition in the text.

(6) How to calculate the transmittance (i.e., the expression) should be described in the text.

Q10. Please check the term "transmission spectrum" used in the text while "transmittance spectrum" used in the figure legend. What is the difference between them.

Q11: The references used in the text should be improved.

(1) The relevant references of latest literatures regarding on the fabrication issue should be cited in the suitable place of the text.

(2) A few relevant references need to be cited in this article to enrich the background of the mechanism of plasmonic MIM waveguides (e.g., Results in Physics, 17, 103116 (2020)) and plasmonic sensing features (e.g., Journal of Nanoparticle Research 20, 190 (2017)). The above-mentioned references should be cited in the suitable place of the text.

Q12: I suggest to check the typo error and abbreviation (e.g., F-P cavity should have a definition for the first use in the text) throughout the manuscript.

Reviewer 3 Report

It is stated that a novel refractive index sensor is proposed in this article, which can be applied to detect the blood health condition. But it is not clear from the paper, how such a sensor can be designed. If the blood should be placed in the waveguide, how it can be possible, when the diameter of the waveguide is 50 nm and the size of blood cells is several micrometers? Besides, the characteristics of the sensor were calculated for refractive index of air (1.0) and wavelength in the IR region of the spectrum, while in the considered example the dependence of RI on the concentration of hemoglobin shows that RI of a blood is more than 1.34 and it is plotted for visible range of spectrum.

I think, the authors need to focus on the properties of the MIM structure itself. Because it is not shown how the other parameters apart from R1, d and h influence on the properties of the structure. It is not clear why the crucial parameter is R1 and not r1, r2 or R2 or w. Also, if there is inverse dependence of FOM on h, which value of h would be optimal? Why were not considered h less than 40 nm?

And, finally, the formula (4) does not describe the relationship between the refractive index and the blood group. It shows the relationship between the refractive index and wavelength.

Round 2

Reviewer 2 Report

This manuscript can now be accepted.